# Latent Space Energy-based Neural ODEs

**Sheng Cheng** †                                                                 *scheng53@asu.edu*
*School of Computing and Augmented Intelligence, Arizona State University*

**Deqian Kong** †                                                                 *deqiankong@ucla.edu*
*Department of Statistics and Data Science, University of California, Los Angeles*

**Jianwen Xie**                                                                   *jianwen@ucla.edu*
*Akool Research*

**Kookjin Lee**                                                                   *kookjin.lee@asu.edu*
*School of Computing and Augmented Intelligence, Arizona State University*

**Ying Nian Wu** ‡                                                               *ywu@stat.ucla.edu*
*Department of Statistics and Data Science, University of California, Los Angeles*

**Yezhou Yang** ‡                                                                 *yz.yang@asu.edu*
*School of Computing and Augmented Intelligence, Arizona State University*

**Reviewed on OpenReview:** *https://openreview.net/forum?id=hCxtlfvL22*

## Abstract

This paper introduces novel deep dynamical models designed to represent continuous-time sequences. Our approach employs a neural emission model to generate each data point in the time series through a non-linear transformation of a latent state vector. The evolution of these latent states is implicitly defined by a neural ordinary differential equation (ODE), with the initial state drawn from an informative prior distribution parameterized by an Energy-based model (EBM). This framework is extended to disentangle dynamic states from underlying static factors of variation, represented as time-invariant variables in the latent space. We train the model using maximum likelihood estimation with Markov chain Monte Carlo (MCMC) in an end-to-end manner. Experimental results on oscillating systems, videos and real-world state sequences (MuJoCo) demonstrate that our model with the learnable energy-based prior outperforms existing counterparts, and can generalize to new dynamic parameterization, enabling long-horizon predictions.

## 1 Introduction

Top-down dynamic generators, such as Pathak et al. (2017); Tulyakov et al. (2018); Xie et al. (2019; 2020), are the predominant models for representing and generating high-dimensional, regularly-sampled, discrete-time sequences, such as text and video. However, they are less suited for irregularly-sampled, continuous-time sequences, commonly found in medical datasets (Goldberger et al., 2000), physical science (Karniadakis et al., 2021). Neural ordinary differential equations (ODEs) (Chen et al., 2018) have emerged as a great approach for the continuous representation of time sequences, demonstrating impressive results in synthesizing new continuous-time sequences. Neural ODEs model describe hidden state dynamics using an ODE defined by a neural network, which takes the current state and time-step as input and outputs the derivative. Given an initial state, a neural ODE defines a continuous-time trajectory of hidden states through numerical ODE solver. As Chen et al. (2018) has shown, the solver's gradient with respect to the neural network parameters

---

† Equal contribution, ‡ Equal advising

can be efficiently computed, enabling neural ODEs to serve as building blocks in advanced deep learning frameworks.

Neural ODEs have been adapted for continuous-time sequences through approaches like ODE-RNNs and Latent ODEs (Rubanova et al., 2019). ODE-RNNs generalize RNNs with continuous-time hidden dynamics, while Latent ODEs (Chen et al., 2018; Rubanova et al., 2019) extend latent variable sequential models to continuous dynamics. They assume the initial state follows an isotropic Gaussian distribution as the prior and recruit an additional inference network for variational inference (Kingma & Welling, 2014) during training.

However, latent ODEs face three fundamental challenges: (1) The design complexity of the inference network, which must effectively capture continuous-time hidden dynamics. (2) The optimization challenges stemming from the evidence lower bound, which introduces a non-zero Kullback–Leibler (KL) divergence between the true posterior and the approximated Gaussian posterior. (3) The limitations of simple Gaussian priors in capturing the complexity of initial latent state spaces.

To address these challenges comprehensively, we introduce ODE-LEBM, a novel latent space neural-ODE based dynamic generative model for continuous-time sequences. Our solution tackles each challenge through specific design choices: (1) To eliminate the inference network complexity, we employ an empirical Bayes approach with Markov chain Monte Carlo (MCMC)-based inference, directly sampling from the posterior distribution without requiring a separate inference network. This approach maintains statistical rigor while simplifying the model architecture. (2) To resolve the optimization challenges from the evidence lower bound, we utilize maximum likelihood estimation (MLE) combined with MCMC sampling. This eliminates the KL divergence gap present in variational approaches, as we directly optimize the likelihood using samples from the true posterior distribution rather than an approximated one. (3) To address the limitations of simple Gaussian priors, we incorporate an energy-based model (EBM) prior (Pang et al., 2020) for the initial ODE state. This significantly enhances the model's expressiveness in capturing complex initial state distributions.

Our training process unifies these solutions through MCMC sampling of latent initial states from both EBM prior and posterior distributions. We update the EBM priors based on the statistical difference between these samples, while the ODE-based generator is updated using posterior samples and observed data.

Our extensive experiments on irregular time series, rotating MNIST, bounding balls, MuJoCo demonstrate that the proposed ODE-LEBM, coupled with an MCMC-based learning algorithm, effectively eliminates the need for complex inference network design, enabling robust capabilities in interpolation and extrapolation beyond the training distribution, and discovering interpretable representations through its expressive EBM priors.

Our work makes the following contributions:

1. We introduce the latent space energy-based neural ODE (ODE-LEBM), a novel ODE-based dynamic generative model with an energy-based prior, designed for continuous-time sequence generation.

2. We train the ODE-LEBM model integrating MLE combined with MCMC-based inference. This training method is principled, statistically rigorous and eliminates the need for inference networks.

3. We present two variants of ODE-LEBM that can discover static and dynamic latent variables in complex systems, improving the model's generalizability and interpretability.

4. We conduct extensive experiments to assess the effectiveness and performance of the proposed ODE-LEBM model and the associated learning algorithms. These experiments evaluate the model's ability to generate realistic continuous-time sequences, learn disentangled and interpretable representations, and outperform existing approaches.

## 2 Related Work

**Neural ODEs**  Neural ODEs (Chen et al., 2018) aim to model continuous-time sequential data by using neural networks to parameterize the continuous dynamics of latent states. Rubanova et al. (2019) and Brouwer et al. (2019) propose continuous-time RNNs by incorporating neural ODE dynamics. Rubanova et al. (2019)

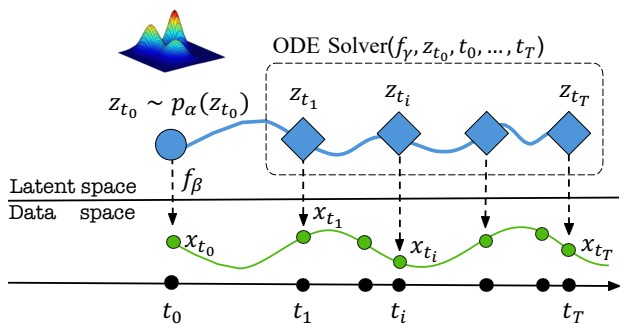

Figure 1: An illustration of ODE-LEBM. The initial latent state $z_{t_0}$ follows a learnable EBM prior distribution $p_\alpha(z_{t_0})$ (Eq. (1)). Subsequent latent states $(z_{t_1}, \ldots, z_{t_T})$ are generated using a neural ODE (Eq. (2)). All latent states are then mapped to the data space through an emission model (Eq. (3)).

and Yildiz et al. (2019) extend the state transition of dynamic latent variable models to continuous-time dynamics specified by neural ODEs and train these models within the VAE framework (Kingma & Welling, 2014). Our work also falls under the category of ODE-based latent variable models. However, unlike Chen et al. (2018); Rubanova et al. (2019); Yildiz et al. (2019); Cheng et al. (2023), our model learns an informative prior for the neural-ODE-based dynamic generator instead of using an uninformative Gaussian prior. Additionally, we train our model via MCMC inference, eliminating the need for an additional assisting network for variational inference.

**Latent space energy-based model** As an intriguing branch of deep energy-based models (Xie et al., 2016; Nijkamp et al., 2019), Latent space EBMs (LEBMs) (Pang et al., 2020) have demonstrated that an EBM can serve as an informative prior model in the latent space of a top-down generator, and can be jointly trained with the generator using maximum likelihood. LEBMs have been successfully applied to a variety of tasks, including text generation (Pang & Wu, 2021; Yu et al., 2022), image generation (Zhu et al., 2023; Cui et al., 2023), trajectory generation (Pang et al., 2021), salient object detection (Zhang et al., 2021; 2023), unsupervised foreground extraction (Yu et al., 2021), molecule design (Kong et al., 2023; 2024a) and offline reinforcement learning (Kong et al., 2024b). From a modeling perspective, Zhu et al. (2023) integrate an LEBM and a neural radiance field (NeRF) (Mildenhall et al., 2022), while Zhang et al. (2021) combines the LEBM with a vision transformer. Cui et al. (2023) incorporate the LEBM into a hierarchical generator, and Yu et al. (2023) pair the LEBM with a diffusion model (Ho et al., 2020). Our work leverages an LEBM alongside an ODE-based generator for modeling continuous-time sequences.

**Dynamical system**. The dynamic texture model (Doretto et al., 2003) employs linear transition and emission models. In contrast, non-linear dynamic generator model has been used to approximate chaotic systems (Pathak et al., 2017), where innovation vectors are given as inputs, and the model is deterministic. The dynamic generator (Tulyakov et al., 2018; Xie et al., 2019; 2020) uses independent innovation vectors at each discrete time step, making the model stochastic. Our model does not incorporate randomness in the dynamical system and is focused on generating continuous-time sequences by leveraging neural ODEs (Chen et al., 2018).

## 3 Latent Space Energy-based Neural ODE

Here, we present our method. Let $\mathbf{x} = (x_{t_i}, i = 1, ..., T)$ denote a sequence of data points, where $(t_i, i = 1, .., T)$ are their timesteps and $T$ is the sequence length. The time $t_i$ is assumed to be known for the observation $x_{t_i}$.

We present a generative model, ODE-LEBM, to represent continuous-time sequences. The model consists of (1) a latent energy-based prior model for the initial state of the latent trajectory, (2) a neural ODE to describe the dynamics of the latent trajectory, and (3) an emission model that produces the time series from

the latent trajectory. The model is illustrated in Figure 1. To be specific, we have

$$z_{t_0} \sim p_\alpha(z_{t_0}), \tag{1}$$

$$z_{t_i} = z_{t_0} + \int_{t_0}^{t_i} f_\gamma(z(t), t) \, \mathrm{d}t, \tag{2}$$

$$x_{t_i} \sim p_\beta(x_{t_i}|z_{t_i}). \tag{3}$$

The energy-based prior model for the initial state in Eq. (1) can be defined as the exponential tilting of a Gaussian distribution:

$$p_\alpha(z_{t_0}) = \frac{1}{Z} \exp\left(f_\alpha(z_{t_0})\right) p_0(z_{t_0}), \tag{4}$$

where $f_\alpha$ is a multilayer perceptron (MLP) with parameters $\alpha$ and $p_0 \sim \mathcal{N}(0, \sigma^2 I)$ is a known Gaussian distribution. $Z = \int \exp(f_\alpha(z_{t_0})) p_0(z_{t_0}) \mathrm{d}z_{t_0}$ is the intractable normalizing constant. The emission model in Eq. (3) is represented by $p_\beta(x|z_{t_i}) = \mathcal{N}(x; f_\beta(z_{t_i}), \sigma_\epsilon^2 I)$, where $f_\beta$ is a neural network with parameters $\beta$. In Eq. (2), $f_\gamma$ is a neural network parameterized by $\gamma$, which specifies a neural ODE (Chen et al., 2018) given by:

$$\frac{\mathrm{d}z(t)}{\mathrm{d}t} = f_\gamma(z(t), t). \tag{5}$$

Given an initial state $z(t_0) = z_{t_0}$, the trajectory of the latent states $z(t)$ is implicitly defined by the ODE, and can be evaluated at any desired time using a numerical ODE solver:

$$z_{t_1}, z_{t_2}, \ldots, z_{t_T} = \mathrm{ODESolve}(f_\gamma, z_{t_0}, t_0, t_1, \ldots, t_T). \tag{6}$$

Following (Chen et al., 2018), we can treat the ODE solver as a differentiable building block for constructing a more complex neural network system.

For notation simplicity, let $\theta = (\alpha, \beta, \gamma)$ be the parameters of the whole model, and $\phi = (\beta, \gamma)$ be those in the ODE-based generation model. Although each $x_{t_i}$ is generated by its latent state $z_{t_i}$, all $\{z_{t_i}, i = 1, ..., T\}$ are generated by the initial state $z_{t_0}$. We can actually write a summarized form for our model

$$\mathbf{x} = F_\phi(z_{t_0}) + \epsilon, \quad \epsilon \sim \mathcal{N}(0, \sigma_\epsilon^2 I), \quad z_{t_0} \sim p_\alpha(z_{t_0}), \tag{7}$$

where $F_\phi$ composes $f_\gamma$ (i.e., ODESolve) and $f_\beta$.

The marginal distribution of our ODE-LEBM is defined as

$$p_\theta(\mathbf{x}) = \int p_\theta(\mathbf{x}, z_{t_0}) \mathrm{d}z_{t_0} = \int p_\phi(\mathbf{x}|z_{t_0}) p_\alpha(z_{t_0}) \mathrm{d}z_{t_0} = \int \prod_{i=1}^{T} p_\phi(x_{t_i}|z_{t_0}) p_\alpha(z_{t_0}) \mathrm{d}z_{t_0}. \tag{8}$$

Suppose we observe a training set of continuous-time sequences $\{\mathbf{x}_m, m = 1, ..., M\}$. The model can be learned by maximizing the likelihood of the training set. The gradient of the log-likelihood is

$$\nabla_\theta \log p_\theta(\mathbf{x}) = \mathbb{E}_{p_\theta(z_{t_0}|\mathbf{x})}[\nabla_\theta \log p_\theta(\mathbf{x}, z_{t_0})] = \mathbb{E}_{p_\theta(z_{t_0}|\mathbf{x})}[\nabla_\alpha \log p_\alpha(z_{t_0}) + \nabla_\phi \log p_\phi(\mathbf{x}|z_{t_0})], \tag{9}$$

which involves the computation of

$$\mathbb{E}_{p_\theta(z_{t_0}|\mathbf{x})}[\nabla_\alpha \log p_\alpha(z_{t_0})] = \mathbb{E}_{p_\theta(z_{t_0}|\mathbf{x})}[\nabla_\alpha f_\alpha(z_{t_0})] - \mathbb{E}_{p_\alpha(z_{t_0})}[\nabla_\alpha f_\alpha(z_{t_0})], \tag{10}$$

and

$$\mathbb{E}_{p_\theta(z_{t_0}|\mathbf{x})}[\nabla_\phi \log p_\phi(\mathbf{x}|z_{t_0})] = \mathbb{E}_{p_\theta(z_{t_0}|\mathbf{x})}\left[\frac{1}{\sigma_\epsilon^2}(\mathbf{x} - F_\phi(z_{t_0}))\nabla_\phi F_\phi(z)\right]. \tag{11}$$

Estimating the expectations in Eq. (10) and Eq. (11) relies on Markov chain Monte Carlo sampling to evaluate the prior distribution $p_\alpha(z_{t_0})$ in Eq. (4) and the posterior distribution $p_\theta(z_{t_0}|\mathbf{x})$.

We could use Langevin dynamics (Neal et al., 2011) to sample from a target distribution of the form $\pi(z) \propto \exp(-U(z))$ by performing a $\Gamma$-step iterative procedure with a step size $\delta$, starting with an initial value $z^{(0)}$ sampled from a standard normal distribution,

$$z^{(\tau)} = z^{(\tau-1)} - \delta \nabla_z U(z) + \sqrt{2\delta} e^{(\tau)}, z^{(0)} \sim \mathcal{N}(0, I), e^{(\tau)} \sim \mathcal{N}(0, I), \quad \text{for} \quad \tau = 1, ..., \Gamma. \tag{12}$$

For the prior distribution $p_\alpha(z)$, the gradient of the potential function is given by $\nabla_z \log p_\alpha(z) = -\nabla_z f_\alpha(z) + z/\sigma^2$. For the posterior distribution $p_\alpha(z|\mathbf{x}) \propto p(\mathbf{x}, z) = p_\alpha(z)p_\phi(\mathbf{x}|z)$, the gradient of the potential function is given by $\nabla_z \log p_\theta(z|\mathbf{x}) = \nabla_z \left[ ||\mathbf{x} - F_\phi(z)||^2/2\sigma_\epsilon^2 - f_\alpha(z) + ||z||^2/2\sigma^2 \right]$.

Algorithm 1 provides a complete description of the training algorithm for our ODE-LEBM.

---

**Algorithm 1** Learning algorithm for our ODE-LEBM

---

**Input**: (1) Training sequences $\{\mathbf{x}_m\}_{m=1}^M$; (2) Number of learning iterations $J$; (3) Numbers of Langevin steps for prior and posterior $\{\Gamma^-, \Gamma^+\}$; (4) Langevin step sizes for prior and posterior $\{\delta^-, \delta^+\}$; (5) Learning rates for energy-based prior and ODE-based generator $\{\xi_\alpha, \xi_\phi\}$; (6) batch size $n'$.

**Output**: Parameters $\phi = (\beta, \gamma)$ for the neural-ODE-based generator and $\alpha$ for the energy-based prior model

1: Initialize $\alpha$, $\beta$, and $\gamma$.
2: **for** $j \leftarrow 1$ to $J$ **do**
3:     Sample observed sequences $\{\mathbf{x}_m\}_{m=1}^{n'}$
4:     For each $\mathbf{x}_m$, sample the prior $z_i^- \sim p_{\alpha_j}(z)$ using $\Gamma^-$ Langevin steps in Eq. (12) with a step size $\delta^-$.
5:     For each $\mathbf{x}_m$, sample the posterior $z_m^+ \sim p_{\theta_j}(z|\mathbf{x}_m)$ using $\Gamma^+$ Langevin steps in Eq. (12) with a step size $\delta^+$.
6:     Update energy-based prior using Adam with the gradient $\nabla\alpha$ computed in Eq. (10) and a learning rate $\xi_\alpha$.
7:     Update the ODE-based dynamic generator using Adam with the gradient $\nabla\phi$ computed in Eq. (11) and a learning rate $\xi_\phi$.
8: **end for**

---

Algorithm 2 provides a complete description of the test phase algorithm for our ODE-LEBM. In testing, given initial, partial or irregular sampled observations $\mathbf{x}$, we generate the remaining sequences $\mathbf{y}$ as conditional generation $p(\mathbf{y}|\mathbf{x}) = \int p(z|\mathbf{x})p(\mathbf{y}|z, \mathbf{x})dz = \mathbb{E}_{p(z|\mathbf{x})}[p(\mathbf{y}|z, \mathbf{x})]$.

---

**Algorithm 2** Test-phase algorithm for our ODE-LEBM

---

**Input**: (1) Testing sequences $\mathbf{x}$; (2) Numbers of Langevin steps for posterior sampling $\Gamma$; (3) Langevin step sizes for posterior sampling $\delta$; (4) Testing prediction timesteps $T$; (5) Learned parameters $\phi = (\beta, \gamma)$ for the neural-ODE-based generator and $\alpha$ for the energy-based prior model.

**Output**: Prediction sequences $\mathbf{y}$.

1: For each $\mathbf{x}$, sample the posterior distribution of the inital state $z_{t_0} \sim p_\theta(z_{t_0}|\mathbf{x})$ using $\Gamma$ Langevin steps in Eq. (12) with a step size $\delta$.
2: Given $z_{t_0}$, predict the latent states $\{z_{t_i}\}_{i=1}^T$ sequence using Eq. (2) and generate the sequences $\mathbf{y}$ using Eq. (3).

---

Algorithms 1 and 2 provide a general framework for learning ODE-LEBM, where latent variables serve as initial latent states ($z_{t_0}$) using MCMC-based inference. While our generator model incorporates a neural ODE in the latent space (Eq. (2)) and an emission model that maps latent states to the data space (Eq. (3)), we aim to enhance model interpretability through a more explicit design of latent variables in above generation process. Specifically, we introduce additional latent variables to disentangle trajectory-specific features in both the neural ODE (Section 4.3.2) and emission procedure (Section 4.3.1). The additional latent variable in the neural ODE, which we refer to as the dynamic variable ($z_d$), captures the hyper-parameters of dynamic evolution that cannot be represented by initial states $z_{t_0}$. Similarly, the additional latent variable in the emission model, termed the static variable ($z_s$), characterizes global environmental features that are not directly involved in dynamic evolution. Both model specifications can be learned using Algorithm 1, leveraging the flexibility of MCMC-based learning and inference algorithms.

# 4 Experiments

This section investigates the performance and adaptability of our proposed ODE-LEBM in various scenarios. By incorporating an EBM prior with an explicit probability density, ODE-LEBM provides a more expressive generative model compared to latent ODE. In the experiments, we evaluate the model's ability to capture complex dynamics from irregularly sampled sequences, discover different time-invariant variables, detect out-of-distribution samples, and demonstrate its applicability in real-world scenarios.

**Baseline methods**   All our baseline methods use neural ODEs as the dynamical model. Their primary difference is in how they encode the latent space variables. Latent ODE (Chen et al., 2018) encodes the initial latent space using an RNN. To address the issue of irregular sampling, Rubanova et al. (2019) proposes the ODE-RNN as an encoder. Additionally, to identify time-invariant variables, Modulated Neural ODEs (MoNODEs) (Auzina et al., 2024) employs an additional inference network for encoding.

**Implementation details**   The network architectures, training details and physical dynamics of the dataset are shown in the Appendix A.

## 4.1 Overview

Our empirical study adopts the convention from Neural ODEs and test our model in the following scenarios.

**Irregularly-sampled time series**   We generate a synthetic dataset of 1000 one-dimensional trajectories, with each containing $T = 100$ irregularly sampled time points within the interval $[0, 5]$. The trajectories are generated using a sinusoidal function with a fixed amplitude of 1, frequency randomly sampled from $[0.5, 1]$, and starting points sampled from $\mathcal{N}(\mu = 1, \sigma = 0.1)$. The dataset is split into 80% for training and 20% for testing. The data generation setting follows Rubanova et al. (2019). During training and testing, the latent variable is inferred using $\gamma T$ randomly sampled timesteps from the total $T$ timesteps to predict all $T$ samples, where $\gamma$ represents the observation ratio (0.1, 0.2, 0.3, or 0.5).

**Rotating MNIST**   The data is generated following the implementation by Casale et al. (2018); Auzina et al. (2024), with the total number of rotation angles set to 16. We include all ten digits from MNIST dataset, with the initial rotation angle sampled from all possible angles $\theta \sim \{0°, 24°, \dots, 312°, 336°\}$. The training data consists of $N = 1000$ trajectories, each with a length of $T = 15$, corresponding to one cycle of rotation. The validation and test data consist of $N_{\text{val}} = N_{\text{test}} = 100$ trajectories, each with a sequence length of $T_{\text{val}} = 15$ and $T_{\text{test}} = 45$. During training and validation, latent variables are inferred from observations over $T$ timesteps to predict within the same interval. For testing, we use the first $T$ timesteps to predict a sequence of length $T_{\text{test}}$. This process involves both interpolation (first $T$ timesteps) and extrapolation (remaining timesteps).

**Bouncing balls with friction**   We use a modified version of the bouncing ball video sequences, a benchmark commonly employed in temporal generative modeling (Gan et al., 2015; Yildiz et al., 2019; Sutskever et al., 2008). The dataset is generated using the original implementation by Sutskever et al. (2008), with an additional friction term sampled from $U[0, 0.1]$ added to each trajectory. The dataset consists of 1000 training sequences of length $T = 20$, and 100 validation and 100 test trajectories with lengths of $T_{\text{val}} = 20$ and $T_{\text{test}} = 40$, respectively. This setup allows for a comprehensive evaluation of the model's ability to learn and predict the dynamics of the bouncing ball under varying friction. The data generation setting follows the implementation by Auzina et al. (2024). The training, validation, and testing protocol follows the same procedure described for Rotating MNIST.

**MuJoCo physics simulation**   MuJoCo physics simulation (Todorov et al., 2012), widely used for training reinforcement learning models, is employed in our experiments with three physical environments: *Hopper*, *Swimmer*, and *HalfCheetah* using the DeepMind control suite (Tunyasuvunakool et al., 2020). We sampled 10,000 trajectories for each environment with 100 timesteps each, with initial states randomly selected. The data was split into training and testing sets at an 80/20 ratio. During training, the sequence length is set to

$T = 50$, while for testing, the sequence length is extended to $T_{\text{test}} = 100$. The training and testing protocol follows the same procedure described for Rotating MNIST.

## 4.2 Irregularly-Sampled Time Series

First, we aim to verify that our proposed ODE-LEBM, along with its associated posterior inference and learning algorithm, performs effectively. Among various choices in the sinusoidal function dataset, those with irregular timesteps pose the greatest challenge. Rubanova et al. (2019) claims that RNN-based inference networks struggle to model time series with non-uniform intervals, proposing ODE-RNNs as an alternative to handle arbitrary time gaps between observations. Instead, we propose using an MCMC-based posterior inference method to avoid the intricate design of inference networks for learning latent ODEs. By incorporating the EBM prior, we are interested in the model's capability to handle irregularly sampled time series.

In training, we randomly subsample a small percentage of time points to simulate sparse observations. For evaluation, we measure the mean squared error (MSE) on the full time series, testing the learned model's interpolation capability. Experiments in Table 1 show the mean squared error for models trained on different percentages of observed points ranging from 0.1 to 0.5. The visualizations are shown in Figure 2. Our ODE-LEBM outperforms Latent ODE with both the RNN encoder and ODE-RNN variant, as well as NCDSSM (Ansari et al., 2023), which is a state space model designed for irregular sampling, validating that ODE-LEBM with MCMC-based posterior inference achieves better performance while eliminating the complexity of designing an inference network. Similarly to Latent ODEs, ODE-LEBMs reconstruct the time series pretty well even with sparse observations and improve performance with increasing observations.

Table 1: Test Mean Squared Error (MSE) on the irregularly-sampled sinusoidal dataset.

| Observation ratio $\gamma$ | 0.1 | 0.2 | 0.3 | 0.5 |
|---|---|---|---|---|
| GRU-D | 2.3562 | 0.0599 | 0.0483 | 0.0411 |
| RNN-VAE | 0.4200 | 0.4189 | 0.4180 | 0.4160 |
| NCDSSM | 0.1710 | 0.1105 | 0.1072 | 0.1029 |
| Latent ODE (RNN enc) | 0.1524 | 0.0374 | 0.0311 | 0.0327 |
| Latent ODE (ODE-RNN enc) | 0.0713 | 0.0311 | 0.0303 | 0.0246 |
| ODE-LEBM | **0.0700** | **0.0304** | **0.0266** | **0.0233** |

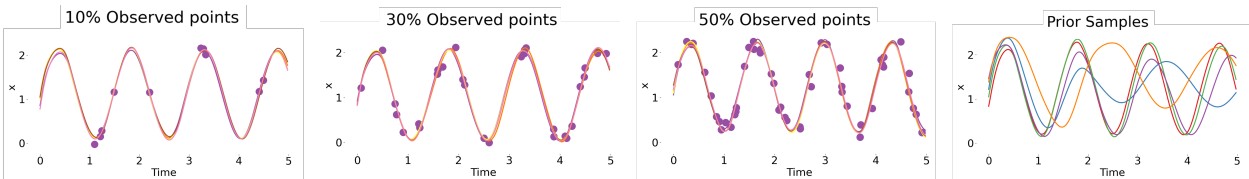

Figure 2: Interpolation results on irregularly-sampled time series. The latent initial state is sampled based on partial observations, $z_{t_0} \sim p_\theta(z_{t_0}|\mathbf{x})$, and then used to predict the entire sequence. For comparison, we also show the results of directly sampling from the learned latent EBM prior, $z_{t_0} \sim p_\alpha(z_{t_0})$ in the last column.

## 4.3 Disentangle Trajectory-specific Latent Variables

Discovering variables that steer the evolution of underlying dynamics in complex dynamical systems can improve generalization to new dynamics and enhance interoperability. Latent variable modeling facilitates the learning of underlying dynamics, particularly when observations and dynamics reside in different spaces, such as in a video recording of a moving car on the road.

The top-down latent variable model (i.e., our ODE-LEBM) can be viewed as a specifically designed model for trajectory-specific variable discovery. The EM-type MLE learning algorithm updates the model parameters using the gradient $\mathbb{E}_{p_\theta(z|\mathbf{x})}[\nabla_\theta \log p_\theta(\mathbf{x}, z)]$ in Eq. (9). The learning process consists of two steps: (1) Infer the latent variable $z \sim p_\theta(z|\mathbf{x})$ from each trajectory using MCMC. The Langevin sampling of the posterior

distribution can be viewed as a gradient-based learning of the trajectory-specific parameters, i.e., $z$. (2) Update the model parameters using $\nabla_\theta \log p_\theta(\mathbf{x}, z)$, where all trajectories share the same model parameters $\theta$.

To further enhance model generalization, we disentangle the latent variable into dynamic latent variables $z_d$, which play a role in ODE evolution, and static variables $z_s$, which capture environment statistics unrelated to dynamics. For simplicity, we concatenate these variables with the initial latent state $z_{t_0}$ and let them share the joint energy-based prior distribution. Unlike MoNODEs (Auzina et al., 2024), which infer deterministic dynamic and static variables using the average of the observation embeddings, our approach infers both latent variables based on their posterior distribution using Langevin dynamics.

### 4.3.1 Trajectory-specific static variables

The first variant of our proposed ODE-LEBM includes a time-invariant static variable $z_s$ to capture variations not directly related to dynamics. $z_s$ plays a role in the emission model, where $x_{t_i} \sim p_\beta(x_{t_i}|z_{t_i}, z_s)$. We assume $z \sim p_\alpha(z)$, where $z = [z_{t_0}, z_s]$, and $[\cdot]$ denotes concatenation. The model, learned by MLE using Algorithm 1, is summarized as follows:

$$[z_{t_0}, z_s] \sim p_\alpha([z_{t_0}, z_s]), \tag{13}$$

$$z_{t_i} = z_{t_0} + \int_{t_0}^{t_i} f_\gamma(z(t), t)\mathrm{d}t, \tag{14}$$

$$x_{t_i} \sim p_\beta(x_{t_i}|z_{t_i}, z_s). \tag{15}$$

We investigate our model's capability on the rotating-MNIST dataset in scenarios where sequence dynamics remain the same but observations differ due to varying static variables in the environment. We explore the following aspects: (1) Interpolation and extrapolation with the learned model; (2) Interpretability of both $z_{t_0}$ and $z_s$; (3) Learned energy functions for out-of-distribution (OOD) detection.

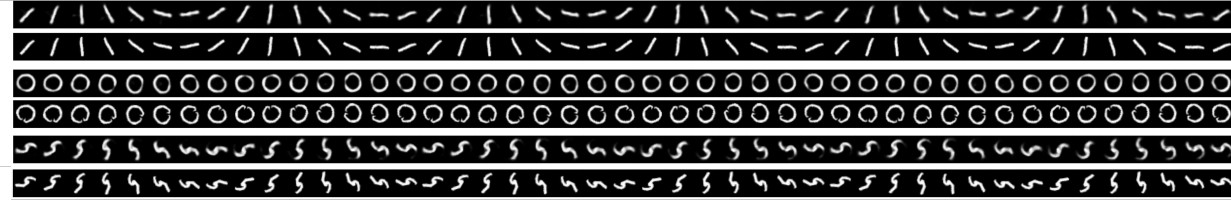

Figure 3: Interpolation and extrapolation results on the rotating-MNIST test set. The first 15 steps represent interpolation, while the last 30 steps represent extrapolation. The first row shows the model predictions for each digit, and the second row presents the ground-truth observations.

Table 2: Mean Squared Error (MSE) on Rotating-MNIST.

|  | T=15 | T=45 | NMI |
|---|---|---|---|
| Latent ODE | 0.015(0.001) | 0.039(0.001) | - |
| LatentApproxSDE | 0.086(0.003) | 0.099(0.003) | - |
| Neural ODE Processor | 0.032(0.004) | 0.035(0.002) | - |
| Modulated ODE | 0.027(0.002) | 0.030(0.001) | 0.131 |
| ODE-LEBM | **0.020**(0.000) | **0.024**(0.001) | **0.576** |

**Interpolation and Extrapolation** To ensure a fair comparison with the baseline model (Auzina et al., 2024), we set the length of the sequences to 15 during training. For evaluation, we measure the MSE on test sequences with lengths of 15 and 45 to demonstrate our model's ability in both interpolation and extrapolation tasks. The interpolation performance is assessed on sequences of the same length as the training data, while the extrapolation performance is evaluated on three times longer sequences. Figure 3 visualizes the model predictions for sequences with 45 time steps, showcasing the model's ability to generate accurate predictions beyond the training sequence length. The experimental results, presented in Table 2, indicate that our model outperforms MoNODEs, which encode static variables with an additional convolutional neural network, in

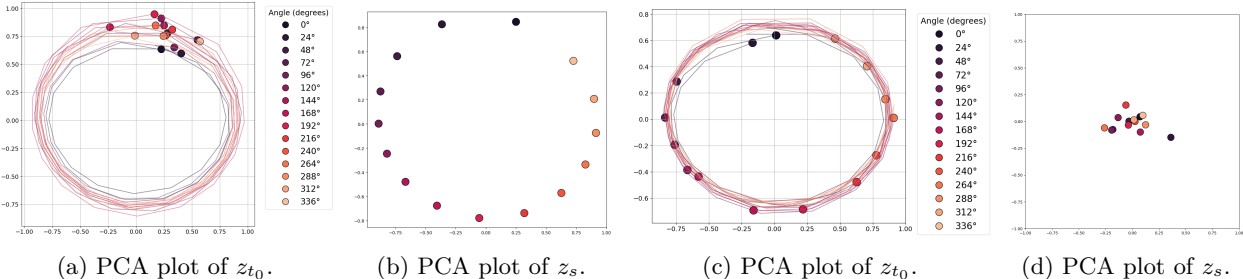

(a) PCA plot of $z_{t_0}$.      (b) PCA plot of $z_s$.      (c) PCA plot of $z_{t_0}$.      (d) PCA plot of $z_s$.

Figure 4: PCA embeddings of both $z_{t_0}$ and $z_s$ for the Rotating MNIST dataset, as inferred by posterior sampling. We generate 16 trajectories from a single digit, incrementing the initial angle of each trajectory by $24°$, starting from $0°$ until $360°$. In (a) and (c), circles denote the start of the trajectory (the initial angle), and lines represent the ODE trajectory. The color gradient corresponds to the initial angle of the trajectory in the observation space. (a) and (b) are from the same sequences with time reset, while (c) and (d) are from the same sequences without time reset.

both interpolation and extrapolation tasks. Additionally, our model outperforms recent approaches like LatentApproxSDE (Solin et al., 2021) and Neural ODE Processor (Norcliffe et al., 2021). These findings demonstrate the effectiveness of the learned energy functions in the latent space, enabling our model to capture the underlying dynamics and generalize well to longer sequences.

**Interpretability of initial states $z_{t_0}$ and static variables $z_s$** To analyze how the model represents rotated digits in its latent space, we conducted experiments by taking a single digit and generating 16 sequences with the length of 16 timesteps each. These sequences show the digit rotating from $0°, 24°, \ldots, 312°, 336°$. For each sequence, we inferred the latent variables ($z_{t_0}$ and $z_s$) using posterior sampling (i.e., $[z_{t_0}, z_s] \propto p([z_{t_0}, z_s]|\mathbf{x})$). To visualize the high-dimensional latent space, we projected these variables into 2D using principal component analysis (PCA).

We investigated two scenarios using the same rotated images but with different time labeling strategies. In the first scenario (Figures 4a and 4b), we reset the timesteps for each sequence to start from 0 to 15, simulating that all sequences begin at $t = 0$. Our results show that $z_{t_0}$ remains constant across all sequences, while $z_s$ captures the rotational information ranging from $0°$ to $336°$. This suggests that by resetting the starting time step, the ODEs focus on the underlying dynamics, which are consistent as they originate from the same digit. In the second scenario (Figures 4c and 4d), we preserved the original timesteps, where each sequence begins at different time points (e.g., first sequence: $t = 0, \ldots, 15$; second sequence: $t = 1, \ldots, 16$; and so on until $t = 15, \ldots, 30$). In this case, $z_{t_0}$ incorporates the rotational information since it needs to account for different starting positions in time, while $z_s$ captures other invariant features of the digit. This experiment demonstrates that the allocation of information between $z_{t_0}$ and $z_s$ is influenced by time-step labeling, even when the underlying rotated images remain unchanged. Furthermore, projecting $z_{t_i}$ into 2D reveals closely aligned trajectories of the 16 sequences, indicating that our model learns consistent dynamics within the dataset, while observation differences are attributed to static variables.

To investigate whether $z_s$ can capture global information, such as the label of the digits, we randomly sample 20 sequences for each digit, with the initial rotation angle randomly set. We project the $z_s$ of each sequence into 2D using t-SNE (Van der Maaten & Hinton, 2008), as shown in Figure 5. The projection reveals sequences corresponding to the same digit cluster together, indicating that $z_s$ successfully encodes the digit label as global information. Furthermore, we compute the Normalized Mutual Information (NMI) of the clustering results to assess the quality of the learned representations quantitatively. MoNODE achieves an NMI of 0.131, while our method obtains a significantly higher NMI of 0.576, demonstrating that our approach produces more interpretable and informative latent variables regarding digit labels.

**Out-of-distribution detection** Energy-based model has been widely used to detect the out-of-distribution samples by treating the energy function as a generative classifier (Elflein et al., 2021). To verify this ability, we generate OOD samples by randomly shuffling the sequence order of real samples. We then visualize the

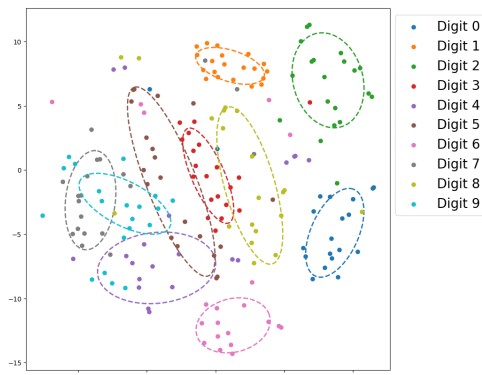

Figure 5: t-SNE plot of static variable $z_s$ with randomly sampled sequences.

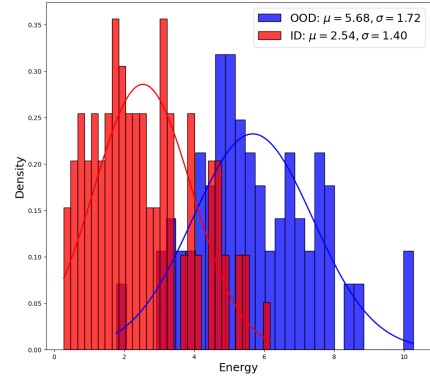

Figure 6: Out-of-distribution detection by energy function.

energy distribution of both real and OOD samples in Figure 6. The visualization shows that OOD samples predominantly fall into the high-energy region, while real samples are primarily located in the low-energy region.

### 4.3.2 Trajectory-specific dynamic variables

We investigate whether latent variables that steer the latent dynamics evolution can emerge with the simplest inductive bias, representing time-invariant hyperparameters in ODEs. We use a dataset of bouncing balls with friction, where each trajectory is sampled from a sequence with different friction coefficients drawn from $U[0, 0.1]$. During training, we observe videos of bouncing balls with length $T = 20$. At test time, we generate sequences with $T = 20$ for interpolation and $T = 40$ for extrapolation. During training and testing, only the first 10 timesteps are used to infer $z_{t_0}$ and $z_d$.

The model, learned by MLE as in Algorithm 1, can be summarized as follows:

$$[z_{t_0}, z_d] \sim p_\alpha([z_{t_0}, z_d]), \tag{16}$$

$$z_{t_i} = z_{t_0} + \int_{t_0}^{t_i} f_\gamma(z(t), z_d, t)\mathrm{d}t, \tag{17}$$

$$x_{t_i} \sim p_\beta(x_{t_i}|z_{t_i}). \tag{18}$$

Table 3: Mean Squared Error (MSE) on bouncing balls with friction (BB).

|  | T=20 | T=40 | $R^2$ |
|---|---|---|---|
| Latent ODE | 0.0178(0.001) | 0.0199(0.001) | -0.29 |
| Modulated ODE | 0.0110(0.000) | 0.0164(0.001) | 0.58 |
| ODE-LEBM | **0.0099**(0.001) | **0.0159**(0.001) | **0.62** |

Table 3 shows the MSE on the test set, demonstrating that our model with dynamic latent variables outperforms other ODE-based counterparts. This highlights the effectiveness of incorporating trajectory-specific dynamic variables to make accurate predictions.

### 4.4 Real-world application: MuJoCo physics simulation

We demonstrate that our ODE-LEBM can be adapted to complex systems like MuJoCo simulation (Todorov et al., 2012). We experiment with three physical environments: *Hopper*, *Swimmer*, and *Cheetah*. To evaluate our method's ability to interpolate and extrapolate in these complex state sequences, we use a subsequence of length 50 during training. At test time, the model observes the first 50 time steps, infers $z_{t_0}$, and predicts

the entire 100 time steps. The first 50 steps resemble the interpolation setting, while the last 50 steps require extrapolation beyond the training distribution. Figure 7 visualizes the prediction results compared to the ground truth. Table 4 shows that ODE-LEBM significantly outperforms latent ODE, demonstrating the capability of our proposed model in real-world applications.

Table 4: Mean Squared Error (MSE) on MuJoCo physics simulation.

|  | Hopper | | Swimmer | | HalfCheetah | |
|---|---|---|---|---|---|---|
|  | T=50 | T=100 | T=50 | T=100 | T=50 | T=100 |
| Latent ODE | 0.0097 | 0.0145 | 5.8e-05 | 6.3e-05 | 0.0112 | 0.0087 |
| ODE-LEBM | **0.0076** | **0.0138** | **3.0e-06** | **4.9e-06** | **0.0081** | **0.0079** |

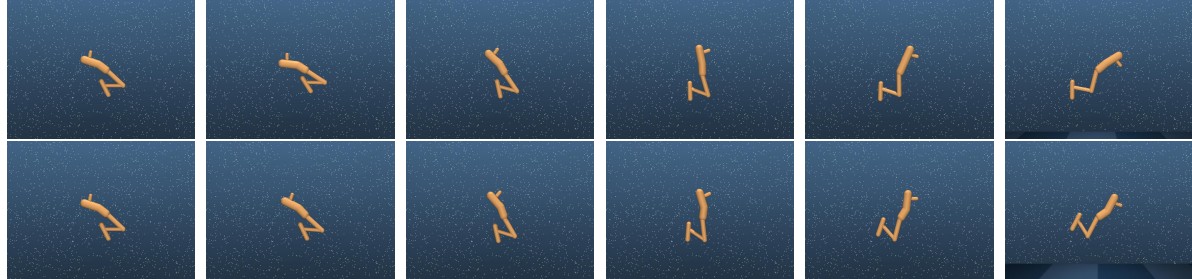

Figure 7: Visualization of prediction results on the MuJoCo *Hopper* environment with time interval 20 steps. The upper row shows the ground truth, while the lower row presents the model predictions. The first 50 time steps (first three columns) represent interpolation, where the model is conditioned on observed data, and the remaining 50 time steps (last three columns) correspond to extrapolation.

## 4.5  Ablation Studies

Table 5: Ablation study on the number of steps in Langevin sampling.

| Steps | 20 | 50 | 100 | 1000 |
|---|---|---|---|---|
| T=15 | 0.020 | 0.014 | 0.012 | 0.011 |
| T=45 | 0.024 | 0.022 | 0.020 | 0.019 |

**Number of Langevin sampling steps during test**   We conduct the experiment on the rotating MNIST, starting from 20 steps, which are set for training. We show that increasing the number of Langevin steps for the posterior during the test can further improve the performance as shown in Table 5.

**The necessity of Latent EBM prior and MCMC-based inference**   To demonstrate the significance of the EBM prior model and MCMC-based inference in posterior inference, we conducted research using the Spiral2D experiment from Chen et al. (2018). We generate 1000 samples each for training and validation, and 200 samples for testing. The sequence length for training and validation is 100, while the test sequence length is 500. We train four models: Latent ODE, MCMC-based posterior inference without an EBM prior, Latent ODE with an EBM prior, and MCMC-based posterior inference with an EBM prior. The test MSE losses for these models are shown in Table 6. The substantial improvement achieved by using the EBM prior and MCMC-based sampling underscores their importance.

**Time Complexity**   We analyze the computational complexity of ODE-LEBM. The analysis is conducted under the same experimental setup as described in Table 6. We report the training and testing speeds of the latent ODE and ODE-LEBM, considering Langevin sampling steps set to 20 (the default setting for all other experiments), 50, and 100. The results are shown in Table 7.

Table 6: The necessity of Latent EBM prior and MCMC-based inference.

| MCMC-based inference | EBM prior | Test MSE |
|:---:|:---:|:---:|
| ✗ | ✗ | 22.76 |
| ✗ | ✓ | 17.92 |
| ✓ | ✗ | 8.33 |
| ✓ | ✓ | 1.36 |

Table 7: Ablation study on time complexity.

| Methods | Latent ODE | ODE-LEBM (20 steps) | ODE-LEBM (50 steps) | ODE-LEBM (100 steps) |
|:---|:---:|:---:|:---:|:---:|
| Training phase (s/iter) | 0.27 | 4.5 | 10.7 | 21.3 |
| Testing phase (s/iter) | 0.27 | 4.2 | 10.4 | 21.0 |

## 5 Conclusion

This paper presents a novel time series model, which is a latent space energy-based neural ODE (ODE-LEBMs) designed for continuous-time sequences. Our model specifies the dynamics of latent states using neural ordinary differential equations (Neural ODEs), represents the initial state space through a latent energy-based prior model, and maps latent states to data space using a top-down neural network. Through extensive experiments, we demonstrate that our model can effectively interpolate and extrapolate beyond the training distribution, unveil meaningful representations, and adapt to various scenarios, including learning from irregularly sampled sequences and identifying different time-invariant variables. Compared to its counterpart model, Latent ODE, our approach is more statistically rigours, accurate, and design friendly.

**Limitation**   This paper proposes using posterior inference to learn ODE-LEBM. Posterior sampling using Langevin dynamics as a form of test-time computation provides a flexible sampling process at the cost of multi-step model forward and backward computation. In the future, we shall study the potential behavior of sampling using Langevin dynamics as a stochastic differential equation or persistent Markov Chain to speed up the sampling process. In the future, we shall leverage the expressive ODE-LEBM to tackle more challenging large-scale real-world applications.

**Acknowledgments**

SC and YW were supported by NSF Cyber-Physical System (CPS) program grant #2038666 and the Engineering Research and Development Center - Information Technology Laboratory (ERDC-ITL) under Contract No. W912HZ24C0022. KL acknowledges support from NSF under grant IIS #2338909. YW was partially supported by NSF DMS-2015577, NSF DMS-2415226, and a gift fund from Amazon. JX acknowledges support from XSEDE grant CIS210052. The authors acknowledge resources and support from the Research Computing facilities at Arizona State University. The views and opinions of the authors expressed herein do not necessarily state or reflect those of the funding agencies and employers.

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

## A  Experiment details

**Model Architecture**  For the emission model and neural ODE model, the network architecture designs are the same as the baseline model in Auzina et al. (2024); Rubanova et al. (2019). For EBM prior, all experiments use 3 layers MLP with activation GELU. The latent space dimension for each experiment is shown in Table 8.

Table 8: The latent space dimension

| Experiment name | $z_t$ | $z_s$ | $z_d$ |
|---|---|---|---|
| Irregular sampling | 10 | - | - |
| Bouncing balls | 8 | - | 4 |
| Rotating MNIST | 16 | 16 | - |
| MuJoCo | 15 | - | - |

**Training details of prior model and MCMC-based inference**  The number of steps in Langevin sampling is 20 for training. We list the remaining training details in Table 9.

**Computing resources for training**  All experiments can be done within an 11GB GPU like GTX 1080Ti. However, to speed up the training process, we use a single V100 for training.

Table 9: The training details of prior model

| Experiment name | hidden dim | step size | learning rate |
|---|---|---|---|
| Irregular sampling | 25 | 0.01 | 1e-4 |
| Bouncing balls | 20 | 0.5 | 1e-5 |
| Rotating MNIST | 20 | 0.5 | 1e-5 |
| MuJoCo | 25 | 0.001 | 1e-5 |

**Physical Dynamics in Dataset  Bouncing balls with friction:** We utilize the script provided by Sutskever et al. (2008), with a minor modification. Each observed sequence is assigned a friction constant $\gamma$, drawn from a uniform distribution $U[0, 0.1]$. The velocity evolves as $v(t) = v_0 - \gamma t$ until the ball collides with a wall, at which point the direction reverses.

**Rotating MNIST Dataset:** We use the image rotation implementation provided by Solin et al. (2021). The total number of rotation angles is set to $T = 16$, with rotations performed around the center of the image. At each timestep, the image rotates by $\pi/8$.

**MuJoCo Physics**: The dynamics in MuJoCo follow a dynamical system as implemented in Todorov et al. (2012)

# B    More visualization

In this section, we will provide more visualization results.

## B.1    Bouncing balls with friction

We present the reconstruction results over 20 timesteps in Figure 8, demonstrating that our model can almost reconstruct the movement of the ball. In contrast, according to Auzina et al. (2024), the latent ODE is unable to achieve this, and MoNODE's reconstruction is also inadequate.

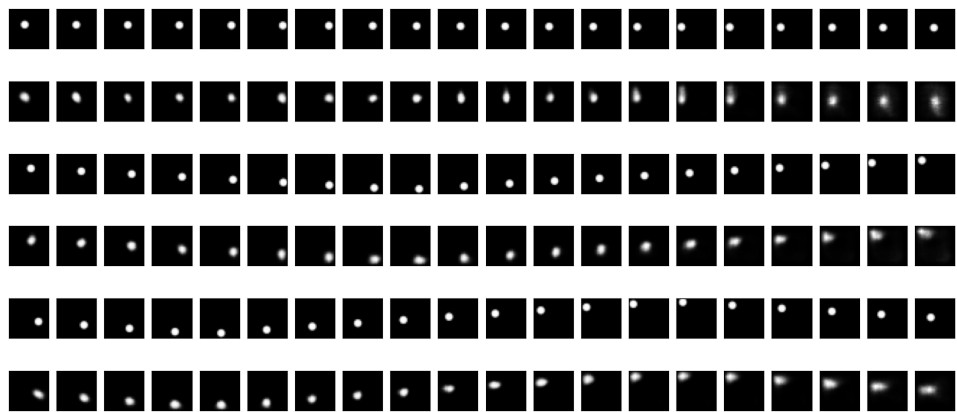

Figure 8: Visualization of prediction results on the Bouncing balls with friction. The first row is the ground truth and the second row is the prediction.

## B.2    Hopper

We present one more pair of Hopper environment results in Figure 9.

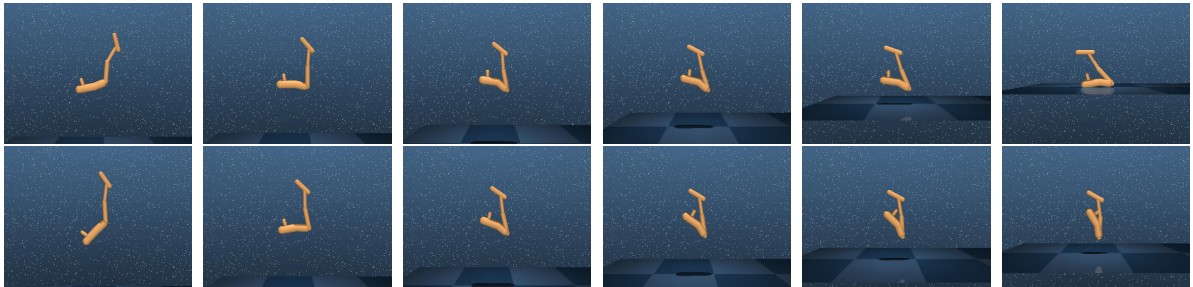

Figure 9: Visualization of prediction results on the MuJoCo Hopper environment. The first row is the ground truth and the second row is the prediction.

## C   Derivations

The derivations of Eqs. (9) to (11) are as follows.

$$
\begin{aligned}
\nabla_\theta \log p_\theta(\mathbf{x}) &= \frac{1}{p_\theta(\mathbf{x})} \nabla_\theta p_\theta(\mathbf{x}) \\
&= \frac{1}{p_\theta(\mathbf{x})} \nabla_\theta \int p_\theta(\mathbf{x}, z_{t_0}) dz_{t_0} \\
&= \frac{p_\theta(z_{t_0}|\mathbf{x})}{p_\theta(\mathbf{x}, z_{t_0})} \nabla_\theta \int p_\theta(\mathbf{x}, z_{t_0}) dz_{t_0} \\
&= \int p_\theta(z_{t_0}|\mathbf{x}) \frac{1}{p_\theta(\mathbf{x}, z_{t_0})} \nabla_\theta p_\theta(\mathbf{x}, z_{t_0}) dz_{t_0} \\
&= \int p_\theta(z_{t_0}|\mathbf{x}) \nabla_\theta \log p_\theta(\mathbf{x}, z_{t_0}) dz_{t_0} \\
&= \mathbb{E}_{p_\theta(z_{t_0}|\mathbf{x})}[\nabla_\theta \log p_\theta(\mathbf{x}, z_{t_0})] \\
&= \mathbb{E}_{p_\theta(z_{t_0}|\mathbf{x})}[\nabla_\alpha \log p_\alpha(z_{t_0}) + \nabla_\phi \log p_\phi(\mathbf{x}|z_{t_0})],
\end{aligned}
\tag{19}
$$

With $p_\alpha(z_{t_0}) = \frac{1}{Z}\exp{(f_\alpha(z_{t_0}))}p_0(z_{t_0})$, and $Z = \int \exp{(f_\alpha(z_{t_0}))}p_0(z_{t_0})dz_{t_0}$,

$$
\begin{aligned}
\mathbb{E}_{p_\theta(z_{t_0}|\mathbf{x})}[\nabla_\alpha \log p_\alpha(z_{t_0})] &= \mathbb{E}_{p_\theta(z_{t_0}|\mathbf{x})}[\nabla_\alpha(\log p_\alpha(z_{t_0}) - \log Z)] \\
&= \mathbb{E}_{p_\theta(z_{t_0}|\mathbf{x})}[\nabla_\alpha \log p_\alpha(z_{t_0})] - \frac{1}{Z}\nabla_\alpha \int \exp{(f_\alpha(z_{t_0}))}p_0(z_{t_0})dz_{t_0} \\
&= \mathbb{E}_{p_\theta(z_{t_0}|\mathbf{x})}[\nabla_\alpha \log p_\alpha(z_{t_0})] - \frac{1}{Z}\int \nabla_\alpha \exp{(f_\alpha(z_{t_0}))}p_0(z_{t_0})dz_{t_0} \\
&= \mathbb{E}_{p_\theta(z_{t_0}|\mathbf{x})}[\nabla_\alpha \log p_\alpha(z_{t_0})] - \int \frac{1}{Z}\exp{(f_\alpha(z_{t_0}))}p_0(z_{t_0})\nabla_\alpha f_\alpha(z_{t_0})dz_{t_0} \\
&= \mathbb{E}_{p_\theta(z_{t_0}|\mathbf{x})}[\nabla_\alpha \log p_\alpha(z_{t_0})] - \int p_\alpha(z_{t_0})\nabla_\alpha f_\alpha(z_{t_0})dz_{t_0} \\
&= \mathbb{E}_{p_\theta(z_{t_0}|\mathbf{x})}[\nabla_\alpha f_\alpha(z_{t_0})] - \mathbb{E}_{p_\alpha(z_{t_0})}[\nabla_\alpha f_\alpha(z_{t_0})].
\end{aligned}
\tag{20}
$$

With $p(\mathbf{x}|z_{t_0}) = \mathcal{N}(F_\phi(z_{t_0}), \sigma_\epsilon^2 I)$,

$$
\begin{aligned}
\mathbb{E}_{p_\theta(z_{t_0}|\mathbf{x})}[\nabla_\phi \log p_\phi(\mathbf{x}|z_{t_0})] &= \mathbb{E}_{p_\theta(z_{t_0}|\mathbf{x})}\left[\nabla_\phi - \frac{1}{2\sigma_\epsilon^2}||\mathbf{x} - F_\phi(z_{t_0})||^2\right] \\
&= \mathbb{E}_{p_\theta(z_{t_0}|\mathbf{x})}\left[\frac{1}{\sigma_\epsilon^2}(\mathbf{x} - F_\phi(z_{t_0}))\nabla_\phi F_\phi(z)\right].
\end{aligned}
\tag{21}
$$

# D  More Quantitative result on Rotating MNIST

**The importance of static variables** $z_s$  We run additional experiment of proposed method without infer the static variables $z_s$. The results are presented in the third row of the Table 10. Our results indicate that our method, without explicitly inferring the static latent variable, achieves better performance than the vanilla latent ODE. However, it does not match the performance of the Modulated ODE, which leverages additional encoders to encode the static latent variable. Notably, when our model infers the static latent variable directly, it outperforms the Modulated ODE.

Table 10: The importance of static variables $z_s$ on Rotating-MNIST.

| Method | Static latent variable $z_s$ | Using Encoder | T = 45 |
|---|---|---|---|
| Latent ODE | ✗ | ✓ | 0.039 |
| Modulated ODE | ✓ | ✓ | 0.030 |
| ODE-LEBM | ✗ | ✗ | 0.036 |
| ODE-LEBM | ✓ | ✗ | 0.024 |

**The necessity of Latent EBM prior and MCMC-based inference**  Similar to the Spiral2D experiment, we conducted an additional ablation study to assess the contributions of the Latent EBM prior and MCMC-based inference, as shown in Table 11.

Table 11: The necessity of Latent EBM prior and MCMC-based inference on Rotating-MNIST.

| Method | MCMC-based inference | EBM prior | T = 45 |
|---|---|---|---|
| Latent ODE | ✗ | ✗ | 0.039 |
| ODE-LEBM | ✓ | ✗ | 0.025 |
| ODE-LEBM | ✗ | ✓ | 0.031 |
| ODE-LEBM | ✓ | ✓ | 0.024 |

