# OpenReview forum: "Latent Space Energy-based Neural ODEs"
_TMLR — Accepted by TMLR_

### Review · Reviewer_rpdn · 2024-12-05

**Summary Of Contributions:**

This paper introduces a novel latent dynamical model in which the latent dynamics are represented by an ODE with an energy-based prior for the initial states. The model is trained using MCMC-based inference to estimate both the prior and posterior distributions of the latent variables. Furthermore, the authors propose two model variants designed to capture global static and dynamic latent variables. Comprehensive experiments are conducted to evaluate the method's performance across various aspects, including its ability to handle irregularly sampled temporal data, interpolate and extrapolate sequential data, disentangle global static and dynamic variables, and apply effectively to real-world datasets.

**Audience:**

Yes

**Broader Impact Concerns:**

No broader impact concerns.

**Claims And Evidence:**

Yes

**Requested Changes:**

I have summarized the requested changes into two parts: theoretical presentation and experimental implementation.

For the theoretical presentation:

- To enhance the clarity of the proposed method, it would be helpful if the authors included the derivation of the learning objectives in equations (10) and (11) in the appendix.

- It would also enhance comprehension if the authors could include the algorithm for inference during the test phase of the proposed model.

For the experimental implementation:

- In Section 4.1, the authors briefly describe the different datasets used for evaluation. However, the underlying true dynamics used to generate three of the datasets—the bouncing balls with friction, rotating MNIST, and MuJoCo physics—are not clearly explained.
Besides, the latent dimensions of $z_t$, $z_s$ and $z_d$ used for experiments are not specified.

- The presentation in Section 4.3.1 on the interpretability of initial states $z_{t_0}$ and static variables $z_s$ is difficult to follow. In particular, it is unclear what it meant by “resetting the timesteps” for a given sequence of observations and how this affects the model’s processing. Providing an example to illustrate this operation would greatly enhance the readers’ understanding.

- In both Figure 6 and Figure 8, it appears that the extrapolation generated by the proposed model tends to produce static motion patterns, with limited variation in object movement. Could the authors provide an explanation for this phenomenon?

**Strengths And Weaknesses:**

Strengths:

- The use of an energy-based prior distribution for the initial states in an ODE-based latent dynamical model, combined with MCMC-based sampling to estimate the expectations with respect to the posterior distribution instead of relying on an encoder, is a novel approach. Extensive experiments conducted by the authors have demonstrated the effectiveness of the proposed method compared to other ODE-based latent dynamical models.

- The proposed method is grounded in solid theoretical analysis with reasonable assumptions.

- The proposed method is evaluated on several different datasets, varying from simple one-dimensional synthetic time series data to physical simulation data. And different aspects of the model are tested. Adequate experimental evidence has proven the superiority of the proposed method.

- The paper is well-structured, clearly presented and easy to follow.

Weaknesses:

Although the paper is concise in its presentation, it lacks certain details in both the theoretical derivations and the experimental implementations. See requested changes for details.

---

> ### Author Response · Authors · 2025-01-03
>
> We greatly appreciate your thorough review . We would like to clarify your concern.
>
> > derivation of the learning objectives and
>
> We add the derivation of the learning objectives in the Appendix C.
>
> > Test-time algorithm
>
> We update the test-phase algorithm of ODE-LEBM in the revision of the paper at the end of Section 3.
>
> > Dataset setting and latent dimension of the model.
>
> Thank you for your suggestions. We have added detailed descriptions of the true dynamics for the three datasets in Appendix A and summarize them here:
> * Rotating MNIST Dataset: We use the image rotation implementation provided by [1]. The total number of rotation angles is set to  $T = 16$ , with rotations performed around the center of the image. At each timestamp, the image rotates by  $\pi/8$ .
> * Bouncing Ball with Friction Dataset: We utilize the script provided by [2], with a minor modification. Each observed sequence is assigned a friction constant  $\gamma$ , drawn from a uniform distribution  $U[0, 0.1]$ . The velocity evolves as  $v(t) = v_0 - \gamma t$  until the ball collides with a wall, at which point the direction reverses.
> * MuJoCo Physics: The dynamics in MuJoCo follow a non-linear system as implemented in [3].
>
> The latent dimensions used in our experiments are as follows:
> | Experiment    | $z_t$| $z_s$ | $z_d$|
> | -------- | ------- |------- |------- |
> | bouncing ball with friction  | 8   | -| 4|
> | Rotating MNIST    | 16    |16 |-|
> | MuJoCo    | 15    |- |-|
>
> > Example of resetting timesteps
>
>
> To analyze how the model represents rotated digits in its latent space, we conducted an experiment using sequences of a single digit rotating over time. Each sequence has a length of 16 timesteps, corresponding to rotation angles ${0^\circ, 24^\circ, \dots, 312^\circ, 336^\circ}$. The latent variables $(z_{t_0}$ and $z_s)$ were inferred for each sequence using posterior sampling $([z_{t_0}, z_s] \propto p([z_{t_0}, z_s] | \mathbf{x}))$.
>
> Scenario 1: Resetting Timesteps
>
> In this scenario, we reset the timesteps for all sequences so that each sequence starts from $t = 0$ and ends at $t = 15$. This simulates the condition where all sequences are aligned to begin at the same time origin. The results show:
> * $z_{t_0}$: Remains constant across all sequences, reflecting that the starting position of the digit does not change when timesteps are reset.
> * $z_s$: Encodes the rotational information of the digit, ranging from $0^\circ$ to $336^\circ$.
>
> This suggests that resetting the timesteps allows the model to focus on the shared underlying dynamics of the sequences, which remain consistent as all sequences represent the same digit.
>
> Scenario 2: Original Timesteps
>
> In this scenario, we preserved the original timesteps, where each sequence starts at a different time point. For example:
> * The first sequence spans $t = 0, \dots, 15$,
> * The second sequence spans $t = 1, \dots, 16$, and so on,
> * Until the last sequence spans $t = 15, \dots, 30$.
>
> Here, the results show:
> * $z_{t_0}$: Encodes rotational information to account for the different starting points of the sequences in time.
> * $z_s$: Captures other invariant features of the digit, unrelated to rotation.
>
> This indicates that the allocation of information between $z_{t_0}$ and $z_s$ depends on the time-step labeling strategy. When timesteps are preserved, $z_{t_0}$ absorbs the rotational information due to the variation in starting positions.
>
>
> > static motion in Figure 6.
>
> This is a common issue of neural ODEs: as time evolves, the total energy of the system tends to drift toward lower-energy states. This observation aligns with findings reported in the Hamiltonian Neural Networks paper by [4]. A potential solution to address this issue would involve improving the dynamics model to more accurately capture and represent the underlying system dynamics.
>
>
> [1] Sutskever, Ilya, Geoffrey E. Hinton, and Graham W. Taylor. "The recurrent temporal restricted boltzmann machine." Advances in neural information processing systems 21 (2008).
>
> [2] Solin, Arno, Ella Tamir, and Prakhar Verma. "Scalable inference in sdes by direct matching of the fokker–planck–kolmogorov equation." Advances in Neural Information Processing Systems 34 (2021): 417-429.
>
> [3] Todorov, Emanuel, Tom Erez, and Yuval Tassa. "Mujoco: A physics engine for model-based control." 2012 IEEE/RSJ international conference on intelligent robots and systems. IEEE, 2012.
>
> [4]Greydanus, Samuel, Misko Dzamba, and Jason Yosinski. "Hamiltonian neural networks." Advances in neural information processing systems 32 (2019).

---

### Review · Reviewer_r7Sf · 2024-12-16

**Summary Of Contributions:**

The authors propose ODE-LEBM, a generative model for continuous-time sequences. It consists of an energy-based model prior for the initial latent state, a neural ODE for the latent trajectory dynamics, and an emission model producing the time series from the latent trajectory. All model components are jointly trained using MLE with MCMC-based sampling.

They conduct experiments on synthetic 1D data, the "bouncing ball video sequences" benchmark, rotating MNIST, and three MuJoCo environments (Hopper, Swimmer, HalfCheetah).

They compare their ODE-LEBM with other neural ODE model variants, achieving quite promising performance overall.

**Audience:**

Yes

**Broader Impact Concerns:**

No concerns.

**Claims And Evidence:**

No

**Requested Changes:**

I think this could be a quite solid paper, it could definitely be relevant for the TMLR audience.

However, I think the current version requires some clarifications, see questions above.

**Strengths And Weaknesses:**

Strengths:
- The paper is quite well-written overall.
- The proposed model is conceptually quite simple, I think it makes intuitive sense to have a learnable prior in the form of an EBM.
- The proposed model seems to perform well overall compared to alternative neural ODE models. The ablation results in Table 6 seem quite strong.





Weaknesses:
- Section 4 is not ideally structured, I found it somewhat difficult to follow some parts.
- The aspect of computational cost (both during training and at test-time) is not really discussed at all.



Questions/suggestions:
- How dos the computational cost scale with the number of Langevin steps, during training and at test-time? For example in Table 5 and Table 6, what is the computational cost of those model variations?
- Seems a bit strange to introduce model variations in Section 4.3.1 (eq. (13) - (15)) and Section 4.3.2 (eq. (16) - (18)), could it not make more sense to describe these in Section 3 instead?
- What would the performance of the original model in eq. (1) - (3) be in Table 2 and Table 3? Are the model variations introduced in Section 4.3.1 and Section 4.3.2, respectively, required for ODE-LEBM to perform well there?
- What model variation is used in Section 4.4? (the model in Section 3, Section 4.3.1 or Section 4.3.2?)
- In Figure 6, is the Hopper falling through the floor in the predicted sequence? Is this not a bit strange? Also, the prediction seems to be quite far away from the ground truth towards the end of the sequence, is this expected? Are there any other types of models which could perform better in this task?
- Could be nice to provide an algorithm also for how the model is used at test-time? This is a bit unclear to me.
- The ablation study in Table 6 is interesting and the results seem quite strong. However, why do you introduce a new dataset/task here which hasn't been mentioned before? And, could this model comparison perhaps be done also on some of the other datasets/tasks?
- Given that the paper is shorter than 10 pages, "Limitations" and "Additional Related Work" could be moved from the appendix to the main text?






Minor things:
- The text formatting next to Figure 4 is a bit strange, with just "region" super close to the caption.
- Section 1, "We train the ODE-LEBM model MLE combined with" --> "We train the ODE-LEBM model using MLE combined with"?
- Section 2, ", and molecule design" --> ", molecule design"?
- Section 2, "a LEBM and a neural radiance field" --> "an LEBM and a neural radiance field"?
- Figure 4 is placed after Figure 5, and is also referred to after Figure 5 in the text? (what's currently called Fig 4 should be called Fig 5?)
- Section 4.3.1, "Energy-based model has been" --> "Energy-based models have been"?
- Section 4.3.1, "generative classifier Elflein et al. (2021)" --> "generative classifier (Elflein et al., 2021)"? Same citation formatting issue also in the first sentence of Section 4.4.

---

> ### Author Response · Authors · 2025-01-03
>
> We sincerely thank you for your detailed feedback and constructive suggestions, which have helped us improve our work. Below, we address each of your points in detail:
>
> > Revision on Section 4 [W1]
>
> Thank you for your suggestions on Section 4. We have revised Section 4.
>
> > Computational Cost [W2, Q1]
>
> We conducted a detailed analysis of the computational complexity of ODE-LEBM under the same experimental setup as Table 6 in the paper. The analysis includes training and testing speeds for the latent ODE and ODE-LEBM, considering Langevin sampling steps set to 20 (default for all other experiments in the paper), 50, and 100. The results are summarized below:
> | Method    | Train (s/iter) | Test (s/iter) |
> | -------- | ------- |------- |
> | Latent ODE  | 0.27   |  0.27|
> | Ours (20 steps) | 4.5    |4.2|
> | Ours (50 steps)   | 10.7    |10.4|
> | Ours (100 steps)   | 21.3    |21.0 |
>
> > Describe model variation in Section 3. [Q2]
>
> Thank you for your suggestion. We revise the paper introducing the model variation at the end of Section 3.
>
> > Performance of original model in Table 2 and 3. [Q3]
>
> We performed an additional experiment using the original model defined by Eqs. (1)–(3), which excludes the static latent variable and the additional encoder. The results, obtained on the rotating MNIST dataset, are presented in the third row of the table below.
> Our results indicate that our method, without explicitly inferring the static latent variable, achieves better performance than the vanilla latent ODE. However, it does not match the performance of the Modulated ODE, which leverages additional encoders to encode the static latent variable. Notably, when our model infers the static latent variable directly, it outperforms the Modulated ODE.
> In the rotating MNIST setting, where the data is collected from 10 digits, relying solely on an inference network may not be sufficient to capture the static latent representation effectively. A manual design of the static latent variable could aid in learning the latent representation more accurately.
> | Method    | Static latent variable |Encoder| T = 45|
> | -------- | ------- |------- |------- |
> | Latent ODE  | N   | Y| 0.039|
> | Modulated ODE | Y     | Y|0.030|
> | Ours    | N    |N |0.036|
> | Ours    | Y    |N|0.024 |
>
>
> > model variation is used in Section 4.4 [Q4]
>
> The model described in Section 3 (eq. 1-3) is used for the experiments in Section 4.4.
>
> > Wired performance on Hopper dataset [Q5]
>
> The performance issues observed in Figure 6 are likely due to insufficient constraints on the state during training and testing, resulting in suboptimal predictions. In the revised manuscript, we have updated the visualizations to more accurately reflect the model’s performance.
>
> The performance drop towards the end of the sequence can be attributed to the extrapolation of dynamics, as the model was trained on sequences of 50 steps but tested on sequences of 100 steps. Additionally, the complex dynamics of the system present challenges for a simple neural ODE. For example, we observed that as time progresses, the system’s total energy tends to drift toward lower-energy states[1]. Addressing these challenges would require more sophisticated design considerations in the dynamics modeling, such as adding constrains in the modelling [2].
>
>
>
> > Test-time algorithm [Q6]
>
> We update the test-phase algorithm of ODE-LEBM in the revision of the paper at the end of Section 3.
>
> > Table 6 results on other dataset. [Q7]
>
> The primary reason for conducting the ablation study on the Spiral2D dataset is to isolate the impact of MCMC-based inference and the EBM prior while minimizing the influence of other model design variations, such as the inclusion of static latent variables in the modeling. The Spiral2D dataset, being a well-established and traditional benchmark in the neural ODE literature, is particularly well-suited for evaluating these specific components due to its simplicity and relevance.
>
> To further validate our findings, we also performed an ablation study on the Rotating MNIST dataset. The results are presented below:
> | Method    | MCMC-based inference |EBM prior| T = 45|
> | -------- | ------- |------- |------- |
> | Latent ODE  | N   | N| 0.039|
> | Ours    | Y    |N |0.025|
> | Ours    | N    |Y |0.031|
> | Ours    | Y    |T|0.024 |
>
> These results demonstrate that both MCMC-based inference and the EBM prior significantly contribute to the performance of our model.
>
> > Paper revision [Q8] and minor issue
>
> Thanks for your suggestion, we have revised the paper based on your suggestions.
>
>
> [1] Greydanus, Samuel, Misko Dzamba, and Jason Yosinski. "Hamiltonian neural networks." Advances in neural information processing systems 32 (2019).
>
> [2] Chen, Ricky TQ, Brandon Amos, and Maximilian Nickel. "Learning neural event functions for ordinary differential equations." arXiv preprint arXiv:2011.03902 (2020).

---

> > ### Comment · Reviewer_r7Sf · 2025-01-11
> >
> > Thank you for the response.
> >
> > - Could you add the second and third results tables from the response above to the paper, at least to the appendix?
> >
> > - Figure 9 in the revised pdf is also quite neat, would it not make sense to move this to the main paper?
> >
> >
> > I have read the other reviews and all responses.
> >
> > The other reviews are also quite positive overall, and I think the authors respond well. They have provided most of the clarifications I requested. If the authors respond to the two questions above in this comment, I will recommend Accept.

---

> > > ### Author Response · Authors · 2025-01-14
> > >
> > > Dear Reviewer,
> > >
> > > Thank you for your positive feedback and valuable suggestions. Following your recommendations, we have added the results to Appendix D of the paper and moved the figure to the beginning of Section 3 in the revised version.
> > >
> > > We greatly appreciate your thoughtful input.
> > >
> > > Best,
> > > Authors

---

> > > > ### Comment · Reviewer_r7Sf · 2025-01-15
> > > >
> > > > Thank you, I will recommend accept.

---

### Review · Reviewer_gDHo · 2024-12-18

**Summary Of Contributions:**

The work presents a time series model that utilizes latent space for neural ODE prediction, and moves back to the data space from latent space using a top-down NN. The authors evaluate their work on multiple benchmarks.

**Audience:**

Yes

**Broader Impact Concerns:**

None.

**Claims And Evidence:**

No

**Requested Changes:**

Please see weakness.

**Strengths And Weaknesses:**

Strengths

- The idea is simple and straightforward.
- The authors evaluate their method on vast experiments.
- Inference network designing is not necessary through this formulation.

Weakness
- While simple, it was quite difficult to go through Section 3. I would encourage authors to add a simple intuitive example in the beginning of Section 3
- Introduction section is not direct enough. I had a hard time connecting the introduction to the contribution. A good revision can fix this.
- Utilizing EBM could introduce additional time complexity. Can the authors do a runtime experiment comparing the models?

---

> ### Author Response · Authors · 2025-01-03
>
> Thank you for your thoughtful review of our paper. We would like to clarify your concern.
>
> > Intutive example in the beginning of Section 3. [W1]
>
> Thanks for your suggestion. We add the intution example in the Appendix D of the paper.
>
> > Revision in introduction. [W2]
>
> Thanks for your suggestion, we revise the introduction of the paper.
>
> > Time complexity of model. [W3]
>
> We analyze the computational complexity of ODE-LEBM. The analysis is conducted under the same experimental setup as Table 6 in the paper. We report the training and testing speeds of the latent ODE and ODE-LEBM.
> | Method    | Train (s/iter) | Test (s/iter) |
> | -------- | ------- |------- |
> | Latent ODE  | 0.27   |  0.27|
> | ODE-LEBM | 4.5    |4.2|

---

### Author Response · Authors · 2025-01-03
**Global Response to Reviewers**

Happy New Year! We sincerely appreciate the constructive feedback provided by the reviewers and the AE for their efforts in managing the review process. We are delighted to see the positive evaluations across various aspects of our work.

The reviewers consistently recognized our paper as well-written and easy to follow (Reviewers r7Sf, rpdn). Reviewers gDHo and r7Sf noted the simplicity and straightforwardness of our proposed method, while Reviewer rpdn highlighted the novelty of our approach. Furthermore, all reviewers acknowledged the comprehensiveness of our experiments across various datasets, showcasing superior performance compared to alternative neural ODE models.

We have addressed each reviewer’s concerns in detail and revised the paper accordingly. Below, we summarize the key changes made in response to their feedback.

> Time complexity of model.

We analyze the computational complexity of ODE-LEBM. The analysis is conducted under the same experimental setup as Table 6 in the paper. We report the training and testing speeds of the latent ODE and ODE-LEBM, considering Langevin sampling steps set to 20 (the default setting for all other experiments in the paper), 50, and 100. The results are shown below table.
| Method    | Train (s/iter) | Test (s/iter) |
| -------- | ------- |------- |
| Latent ODE  | 0.27   |  0.27|
| Ours (20 steps) | 4.5    |4.2|
| Ours (50 steps)   | 10.7    |10.4|
| Ours (100 steps)   | 21.3    |21.0 |

> Test-phase algorithm of ODE-LEBM.

We update the test-phase algorithm of ODE-LEBM in the revised manuscript. The algorithm is now included at the end of Section 3 as Algorithm 2.


> Revision in the paper.

We have revised the paper based on the reviewers’ feedback and summarized the major revisions as follows. All revisions in the paper have been highlighted in blue.

* Added the test-time algorithm at the end of Section 3 (Algorithm 2).
* Added time complexity results in the ablation studies section (Section 4.5, Table 7).
* Moved “Related Work” and “Limitations” to the main paper, as suggested by Reviewer r7Sf.
* Revised Sections 1, 3, and 4 for improved clarity, as suggested by Reviewers gDHo and r7Sf.
* Added the derivation of the learning objective in Appendix C, as requested by Reviewer rpdn.
* Included more details on dataset generation and model settings in Appendix A, addressing Reviewer rpdn’s suggestions.
* Added the intution example of the ODE-LEBM in Appendix D, as requested by Reviewer gDHo.

---

### Decision · Action_Editor_XknW · 2025-01-22

**Recommendation:** Accept as is

**Comment:**

The paper proposes a latent ODE as an state-space like emission model, where a latent ODE is coupled with a decoder. The initial state distribution posterior is inferred through MCMC, which presents a simple training objective. The method requires inference at training-time to establish the initial state distribution for test sequences. The paper also proposes a more flexible initial distribution as a transformed Gaussian.

The reviews generally agree that the idea is sensible and well-presented, and comes with extensive empirical evaluation. One reviewer found results incremental, and the added cost of the method as negative. All reviewers agree on the paper presenting a simple and useful method with good results. The authors improved the submission based on the comments by incorporating studies on model interpretability, cost and by clarifying the presentation wrt claims.

This is a well-written paper that proposes a useful, well-presented, and sufficiently novel method in the domain, that solves some of the problems in earlier methods, while achieving good performance.

**Audience:**

All reviewers agree. The work will be interesting for the ODE machine learning community, and the SOTA results will draw interest.

**Claims And Evidence:**

2 reviewers agree, while 1 reviewer disagrees.

The paper proposes a novel formulation of latent ODEs without using the predominant VAE formulation. Instead the paper models a latent ODE with an emission model. The initial state ditribution is made more flexible with energy-based model, while its distribution is inferred through MCMC, and the training loss simplifies. The paper claims to do away with KL gap of VAE-based formulations; make the initial state more flexible; simplify the initial state inference; discover decomposed latent representations; and outperform competing methods. There is sufficient evidence for the theoretical claims, while the results do show superiority (although incremental) against competing methods on a wide array of experiments.

The paper makes precise claims, and supports them with sufficiently convincing evidence.